# Role of endoscopic ultrasonography in the diagnostic work-up of idiopathic acute pancreatitis (PICUS): study protocol for a nationwide prospective cohort study

Devica S Umans ,[1,2] Hester C Timmerhuis,[2,3] Nora D Hallensleben,[2,4] Stefan A Bouwense,[5] Marie-Paule GF Anten,[6] Abha Bhalla,[7] Rina A Bijlsma,[8] Marja A Boermeester,[9] Menno A Brink,[10] Lieke Hol,[11] Marco J Bruno,[4] Wouter L Curvers,[12] Hendrik M van Dullemen,[13] Brechje C van Eijck,[14] G Willemien Erkelens,[15] Paul Fockens,[1] Erwin J M van Geenen,[16] Wouter L Hazen,[17] Chantal V Hoge,[18] Akin Inderson,[19] Liesbeth M Kager,[20] Sjoerd D Kuiken,[21] Lars E Perk,[22] Jan-Werner Poley,[4] Rutger Quispel,[23] Tessa EH Römkens,[24] Hjalmar C van Santvoort,[3,25] Adriaan CITL Tan,[26] Annemieke Y Thijssen,[27] Niels G Venneman,[28] Frank P Vleggaar,[29] Annet MCJ Voorburg,[30] Roy LJ van Wanrooij,[1] Ben J Witteman,[31] Robert C Verdonk,[32] Marc G Besselink,[9] Jeanin E van Hooft,[33] Dutch Pancreatitis Study Group

For numbered affiliations see end of article.

**Correspondence to**
Dr Devica S Umans;
d.s.umans@amsterdamumc.nl

## ABSTRACT

**Introduction** Idiopathic acute pancreatitis (IAP) remains a dilemma for physicians as it is uncertain whether patients with IAP may actually have an occult aetiology. It is unclear to what extent additional diagnostic modalities such as endoscopic ultrasonography (EUS) are warranted after a first episode of IAP in order to uncover this aetiology. Failure to timely determine treatable aetiologies delays appropriate treatment and might subsequently cause recurrence of acute pancreatitis. Therefore, the aim of the Pancreatitis of Idiopathic origin: Clinical added value of endoscopic UltraSonography (PICUS) Study is to determine the value of routine EUS in determining the aetiology of pancreatitis in patients with a first episode of IAP.

**Methods and analysis** PICUS is designed as a multicentre prospective cohort study of 106 patients with a first episode of IAP after complete standard diagnostic work-up, in whom a diagnostic EUS will be performed. Standard diagnostic work-up will include a complete personal and family history, laboratory tests including serum alanine aminotransferase, calcium and triglyceride levels and imaging by transabdominal ultrasound, magnetic resonance imaging or magnetic resonance cholangiopancreaticography after clinical recovery from the acute pancreatitis episode. The primary outcome measure is detection of aetiology by EUS. Secondary outcome measures include pancreatitis recurrence rate, severity of recurrent pancreatitis, readmission, additional interventions, complications, length of hospital stay, quality of life, mortality and costs, during a follow-up period of 12 months.

### Strengths and limitations of this study

► This is the first prospective cohort study of only patients with a single episode of presumed idiopathic acute pancreatitis.
► This is the first prospective cohort study which only includes patients after complete standard diagnostic work-up (including exclusion based on blood serum alanine aminotransferase and imaging after clinical recovery).
► The multicentre nature of this study reduces the risk of patient selection bias.
► This study has a follow-up time of a year, and thus this study could elucidate the previously hypothesised association between endoscopic ultrasonography (EUS), detection of aetiology and subsequent treatment of aetiology, and pancreatitis recurrence.
► As the timing of the EUS is set to be after clinical recovery from pancreatitis in this trial, no conclusions on the diagnostic yield of EUS in a different time frame can be drawn from this study.

**Ethics and dissemination** PICUS is conducted according to the Declaration of Helsinki and Guideline for Good Clinical Practice. Five medical ethics review committees assessed PICUS (Medical Ethics Review Committee of Academic Medical Center, University Medical Center Utrecht, Radboud University Medical Center, Erasmus Medical Center and Maastricht University Medical Center).

The results will be submitted for publication in an international peer-reviewed journal.
**Trial registration number** Netherlands Trial Registry (NL7066). Prospectively registered.

## BACKGROUND

Acute pancreatitis can be induced by numerous causes. Gallstone disease (approximately 50%) and alcohol (approximately 20%) are the most frequent causes,[1–6] although the prevalence of aetiologies of acute pancreatitis is dependent on, among other things, age and geographical factors.[7–10] There is, however, a considerable group of patients of approximately 25% in whom no aetiology can be found after routine diagnostic work-up (ie, medical history, laboratory investigations and transabdominal ultrasound). These patients are considered to have presumed idiopathic acute pancreatitis (IAP).[3]

When IAP is presumed, guidelines recommend repeat transabdominal ultrasound after discharge.[11 12] This repeat ultrasonography has an additional diagnostic yield of 20% for the detection of gallstones or sludge in these patients.[13] Undetected microlithiasis and biliary sludge are generally considered to be the major cause of presumed IAP.[14 15] Undetected and subsequently untreated gallstone disease poses a risk for recurrent acute pancreatitis and other biliary events, for example, cholecystitis, biliary colic and cholangitis.

Therefore, when previous diagnostics failed to uncover an aetiology, endoscopic ultrasonography (EUS) should be considered for the detection of biliary disease or other abnormalities causing pancreatitis, such as neoplasms and chronic pancreatitis.[11 12 16 17] EUS is advised as the first step in presumed IAP, followed by (secretin-enhanced) magnetic resonance cholangiopancreaticography (MRCP) to identify rare morphological abnormalities,[11] as EUS is considered to have a higher diagnostic yield than MRCP for clinically relevant causes.[18]

Although guidelines do recommend performing EUS after a first or second attack of presumed IAP, this recommendation is scored as a mere grade 2C, according to the Grading of Recommendations Assessment, Development and Evaluation classification[19] (indicating a weak recommendation based on evidence of low quality, with weak agreement among experts in this field).[11] Therefore, EUS is not routinely performed as the exact significance in this patient group is unclear.[11 16]

The Pancreatitis of Idiopathic origin: Clinical added value of endoscopic UltraSonography (PICUS) Study was designed to determine whether routine EUS should be incorporated in the standard diagnostic work-up of a first episode of presumed IAP.

## METHODS AND ANALYSIS

### Study aim

The objective of this study is to determine the diagnostic yield of EUS for the detection of aetiology in patients with a first episode of presumed IAP.

Depending on the diagnostic yield of EUS observed in the PICUS Study, incorporation of EUS in routine diagnostic work-up of patients with a first episode of presumed IAP will be considered. A minimal diagnostic yield of 10% for any aetiology will be regarded as reasonable to justify implementing routine EUS in the standard diagnostic work-up of a first episode of presumed IAP.

### Study design and setting

PICUS is a multicentre prospective cohort study. A total of 106 patients will be included from 28 participating Dutch centres, including all eight university centres and 20 large teaching hospitals. A listing of the participating centres is included in the authors' information. An overview of the study design, including screening procedures and follow-up, is provided in figure 1.

### Study population

The subjects of this study have had a first episode of acute pancreatitis, as defined by the 2012 Revised Atlanta criteria,[20] with an unknown origin after standard diagnostic work-up, according to the 2013 International Association of Pancreatology/American Pancreatic Association (IAP/APA) evidence-based guidelines on management of acute pancreatitis.[11] The diagnostic modalities that constitute standard diagnostic work-up are listed in table 1 and online supplementary additional file 1. The diagnostic tests as laid out in table 1 are to be performed in all subjects and these tests cannot show any signs of an aetiology in all subjects. Potential aetiologies and their definitions are listed in table 2 and online supplementary additional file 1.

### Eligibility criteria

The inclusion criteria are:

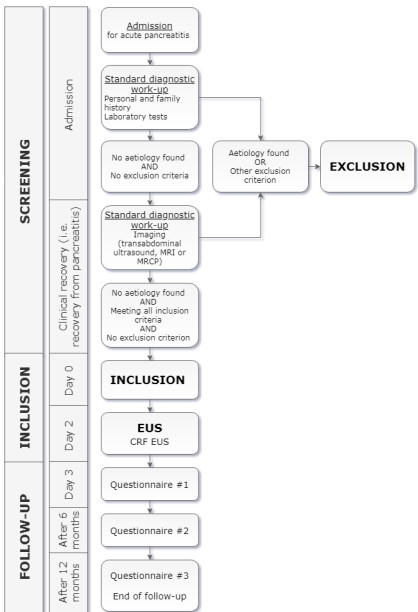

**Figure 1** Overview of screening and study procedures. CRF, Case Report Form; EUS, endoscopic ultrasonography; MRCP, magnetic resonance cholangiopancreaticography.

**Table 1** Standard diagnostic work-up

| Detailed personal and family history, including questions on: | Alcohol use |
| --- | --- |
| | Recent ERCP |
| | Recent start or changes in use of drugs associated with acute pancreatitis |
| | Recent major abdominal trauma |
| | Recent abdominal surgery |
| | Familial and hereditary pancreatitis |
| | Cystic fibrosis-related pancreatitis |
| Laboratory tests, including: | Blood serum triglyceride level |
| | Blood serum calcium level, corrected for the blood serum albumin level |
| | Blood serum ALT level on admission |
| Imaging: | Transabdominal ultrasound, MRI or MRCP after clinical recovery |

Standard diagnostic work-up according to the 2013 International Association of Pancreatology/American Pancreatic Association evidence-based guidelines on management of acute pancreatitis. A listing of the drugs considered to be associated with acute pancreatitis is listed in online supplementary additional file 1.

ALT, alanine aminotransferase; ERCP, endoscopic retrograde cholangiopancreaticography; MRCP, magnetic resonance cholangiopancreaticography.

1. Patients of 18 years or older.
2. First episode of presumed IAP after standard diagnostic work-up, as defined by the IAP/APA evidence-based guidelines on management of acute pancreatitis.[11]
3. Informed consent for participation.
   The exclusion criteria are:
1. Known aetiology.
2. Chronic pancreatitis, as defined by the M-ANNHEIM criteria.[21]
3. Recurrent pancreatitis.
4. Altered anatomy which prohibits the endosonographist from visualising the gall bladder, bile ducts, pancreas or pancreatic duct via EUS (eg, gastric bypass surgery).
5. Diagnostic EUS aimed to determine aetiology before inclusion.

## Endoscopic ultrasonography

EUS will be performed in routine clinical practice by an endosonographist. Use of linear or radial EUS will be at the discretion of the endosonographist. All Dutch endosonographists are trained to perform EUS according to the technique of Hawes and Fockens.[22]

The endosonographist will systematically report, using a standardised Case Report Form (CRF), the experience of the endosonographist, visualisation of anatomical structures (ie, gall bladder, common bile duct and pancreatic duct), presence of local complications of acute pancreatitis, characteristics of biliary aetiology (ie, gallstones,

microlithiasis and/or biliary sludge), characteristics of chronic pancreatitis, presence of (a) pancreatic or peri-ampullary benign or malignant tumour(s), characteristics of auto-immune pancreatitis, anatomic variations (eg, pancreas divisum) or other anomalies (eg, cholecystitis, vascular, renal, splenic or hepatic anomalies or ascites) and performance of fine needle aspiration or fine needle biopsy. Additionally, the type of endoscope, use of sedation, procedure-related complications and results of the fine needle aspiration or biopsy will be systematically recorded by the study coordinator in a separate CRF.

### Primary outcome measure

The primary outcome measure is the number and ratio of patients with presumed IAP in whom EUS detects a cause for the pancreatitis episode.

A positive EUS is defined as an EUS during which a definitive cause for the acute pancreatitis episode has been found or during which abnormalities are visualised constituting a definitive cause, after obtaining tissue and pathological examination. An overview of the exact findings scored as positive imaging is provided in table 3.

If during EUS pancreatic abnormalities are found, yet not enough to make a certain diagnosis of chronic pancreatitis according to the M-ANNHEIM classification,[21] this imaging is considered to be negative, even though it did show abnormalities. This approach is chosen because the aim of this study is to determine the rate of which EUS can find a cause for the presumed IAP episode. For the same reason, report of an anatomical abnormality during EUS after a first episode of acute pancreatitis is not scored as positive imaging as pancreatic morphological changes are very common in IAP and not necessarily clinically relevant, as is elaborated on in the discussion.[23]

### Secondary outcome measures

The secondary outcome measures are recurrence rate of acute pancreatitis, severity of recurrent pancreatitis,[20] readmission, performance of additional invasive procedures (eg, cholecystectomy, endoscopic sphincterotomy), complications of EUS and of additional interventions, according to the Clavien-Dindo classification,[24] length of hospital stay, quality of life, mortality and costs. Relevant definitions are reported in online supplementary additional file 2.

### Sample size calculation

The sample size calculation was based on the primary outcome measure, diagnostic yield of EUS. Based on two previous studies reporting yield in patients with a first episode of presumed IAP,[25 26] adjusted for the PICUS Study criteria for inclusion (ie, requiring negative imaging after clinical recovery) and for positive imaging (ie, excluding pancreas divisum as aetiology), diagnostic yield was assumed to be 30%. Using a two-sided significance level ($\alpha$) of 0.05, a power ($1 - \beta$) of 80%, 95 patients are needed to attain a 95% CI with a range smaller than 10% above and below the assumed yield of 30% (95%

**Table 2** Potential aetiologies and their definitions

| Aetiology | Definition |
|---|---|
| Alcohol | >4 units of alcohol in the 24 hours prior to start of abdominal complaints[51–53] |
| Biliary disease | 1. A transient elevated ALT level of >2 times the upper limit of normal at diagnosis of acute pancreatitis, in the absence of other ALT elevating comorbidity,[34] OR<br>2. Gallstones, microlithiasis and/or biliary sludge, OR<br>3. A dilated CBD of >8 mm in patients <76 years or >10 mm in patients >75 years at diagnosis of acute pancreatitis[36] |
| Cystic fibrosis | History of cystic fibrosis in the absence of another origin[54] |
| Familial | Two or more direct blood-related family members (parents, children or siblings) who have had an episode of acute pancreatitis[55–57] |
| Hereditary | Mutation in the PRSS1, SPINK1, CFTR, CTRC, CLDN2 or CPA1 gene, or direct family member (parents, children, siblings) with one or more of the above mentioned mutations and at least one direct family member who has (had) acute or chronic pancreatitis[57 58] |
| Hypercalcaemia | Blood serum calcium level ≥12 mg/dL (3 mmol/L), corrected for serum albumin level, as first measured during admission[59] |
| Hypertriglyceridemia | Blood serum triglyceride level of ≥1000 mg/dL (11.2 mmol/L) under fasting conditions, as first measured during admission[60] |
| Medication | Use of drug(s) listed in online supplementary additional file 1, which has or have been started or increased in dosage within a reasonable temporal sequence, in principle 1 month before onset of pancreatitis, and has or have a positive dechallenge (a drug reaction that is confirmed by stopping the drug)[61 62] |
| Neoplasm | Known hepatopancreatobiliary malignancy or known malignancy with metastases causing obstruction of the pancreatic duct[63] |
| ERCP | ERCP within 24 hours before diagnosis of pancreatitis[64] |
| Surgical | Abdominal surgery within 24 hours prior to diagnosis of pancreatitis[65] |
| Trauma | Typical blunt trauma to the upper abdomen and pancreatic trauma visible on imaging[66] |

Potential aetiologies and their definitions. Side branch or mixed-type intraductal papillary mucinous neoplasms without dilatation of the pancreatic duct and pancreas divisum will not be considered to be a causative factor for the pancreatitis episode. If imaging is not able to discriminate between gall bladder polyps or concrements, lesions smaller than 10 mm will not be considered an exclusion criterion. Lesions above 10 mm, irrespective of whether they are a polyp or a concrement, are an immediate indication for cholecystectomy, and these patients will be excluded from the " Pancreatitis of Idiopathic origin: Clinical added value of endoscopic UltraSonography" study.
ALT, alanine aminotransferase; CBD, common bile duct; ERCP, endoscopic retrograde cholangiopancreaticography.

CI: 20.8, 39.2). Assuming a drop-out rate of 10%, a total of 106 patients will be included.[27] The sample size was calculated using the software programs RStudio[28] and nQuery.[29]

## Follow-up

Data from patient records on primary and secondary outcome measures will be collected until 1 year after inclusion. Outpatient care and follow-up after the EUS is at the discretion of the treating physician, but an outpatient clinic visit after EUS to discuss the results of the EUS and potential subsequent appropriate treatment can be considered standard care.

In case of biliary disease, the patient will be considered for endoscopic retrograde cholangiopancreaticography (ERCP) with sphincterotomy when choledocho(-micro-)lithiasis or sludge in the common bile duct is present, and cholecystectomy, as is standard care for biliary pancreatitis. A secretin-enhanced MRCP (s-MRCP) will be recommended, if not performed earlier, if a patient is readmitted for a recurrent episode of acute pancreatitis after a negative EUS for aetiology, in order to rule out structural anomalies such as pancreas divisum. This is in accordance with current guidelines.[11]

Patients will be asked to fill out the Short Form-36 Questionnaire in the validated Dutch translation on day 3 after inclusion, after 6 months and after 1 year. This questionnaire in both English and Dutch is included in online supplementary additional file 3.

## Statistical aspects

All included subjects will be evaluated for primary and secondary endpoints until 1 year after inclusion. The primary analysis will be based on intention-to-treat principles. For exploratory reasons a per-protocol analysis will be performed too.

The intention-to-treat population comprises all patients included in the study, regardless of adherence to study protocol. The per-protocol population is the subset of included patients who were treated with the guidelines of the protocol (ie, meeting all eligibility criteria including all of the diagnostic tests required for the diagnosis of IAP, undergoing EUS as described in the Endoscopic ultrasonography section). A tabular listing of all patients excluded from the intention-to-treat population will be provided together with the reasons for exclusion.

All analyses will be performed in the latest available version of SPSS for Microsoft Windows. All data handling

| Table 3 | Positive imaging |
|---|---|
| *Biliary pancreatitis* | Presence of biliary stones, microlithiasis or sludge |
| | Widened CBD, >8 mm in patients <76 years or >10 mm in patients >75 years, in the absence of other CBD dilating factors (eg, opioid use, distal stenosis, obstruction of external compression of CBD or papilla[67]) |
| *Chronic pancreatitis* | Pancreatic calcifications |
| | >4 of the following abnormal features of the pancreas:<br>1. Enlarged gland size<br>2. Cysts<br>3. Echo-poor lesions (focal areas of reduced echogenicity)<br>4. Echo-rich lesions (>3 mm in diameter)<br>5. Accentuation of lobular pattern<br>6. Increased duct wall echogenicity<br>7. Irregularity of the main pancreatic duct<br>8. Dilation of the main pancreatic duct >3.5 mm[68]<br>9. Visible side branches<br>10. Calcifications of the pancreatic duct |
| *Neoplasms* | Definitive diagnosis of pathological tissue after histological or cytological evaluation of specimen of an anomaly observed during EUS, for example, hyperplastic or malignant tissue, or auto-immune inflammatory disease |
| | Main duct IPMN or mixed-type IPMN causing dilatation of the pancreatic duct |

Definition of positive imaging. For each diagnosis, presence of one of the separately mentioned abnormalities is required to be considered as positive imaging. Specimen is not required to be obtained during EUS. Anatomical anomalies (eg, divisum) are not considered a certain aetiology in first episode idiopathic acute pancreatitis and therefore not considered as positive imaging.
CBD, common bile duct; EUS, endoscopic ultrasonography; IPMN, intraductal papillary mucinous neoplasm.

and analysis will be saved in a syntax-file. Results will be presented with all centres combined. A two-tailed p value of <0.05 is considered statistically significant.

### Baseline variables
The reported baseline characteristics consist of age, sex, body mass index (BMI), previous cholecystectomy, nicotine and alcohol use, severity of pancreatitis, length of hospital stay, amylase, lipase, C reactive protein, alanine transaminase, calcium, albumin and triglyceride levels in blood serum on admission, imaging modalities before EUS and their findings. Baseline characteristics of EUS will include timing of EUS, experience of endosonographist and type of sedation and type of endoscope used. Data will be presented in percentages or as mean with SD or in case of a skewed distribution as median with IQR.

### Primary outcome measure: aetiology detection rate
Overall detection rate of an aetiology for the episode of acute pancreatitis will be presented as percentage with a 95% CI. Predefined subgroup analyses will be made for patients with and without obesity (cut-off at a BMI of 30), a previous cholecystectomy, alcohol use and local complications from the IAP episode. A subgroup analysis will also be made for patients with a transabdominal ultrasound as imaging after clinical recovery and with MRI or MRCP as imaging after clinical recovery. Finally, a subgroup analysis will be made for EUS performed by endosonographists with and without extensive experience (cut-off at 400 endosonographies performed), use of linear or radial scope and type of sedation used. In subgroup analyses, the $\chi^2$ test or the Fisher's exact test will be used, as appropriate, to compare aetiology detection rate between

subgroups. In subgroup analyses, comparability between groups regarding baseline variables will be checked. If the subgroups differ statistically significantly in one or more baseline variables, this will be corrected in a logistical regression analysis.

### Secondary outcome measures
Secondary outcome measures will be described as percentages with 95% CI, as mean with SD or median with IQR, as appropriate.

For recurrence rate, subgroup analyses will be made for patients with a positive and negative EUS, and in patients with a positive EUS, for patients who were and were not treated adequately. The same subgroup analyses as in the primary outcome measure will also be applied on the recurrence rate. The $\chi^2$ test or the Fisher's exact test will be used for comparison between subgroups, as appropriate.

For quality of life, subgroup analyses will be made for baseline versus follow-up quality of life and for patients with a positive and negative EUS, and with and without pancreatitis recurrence during follow-up. The (un-) paired t-test, Wilcoxon signed rank test or the Mann-Whitney U test will be used for comparisons between subgroups, as appropriate.

### Cost analysis
The cost analysis will comprise direct medical costs, which are generated by healthcare utilisation and include hospital admission periods and therapeutic and diagnostic procedures.[30] Estimates of unit costs will be based on Dutch reference data from the cost guide of the Dutch Health Council.[31] If this guide is an inappropriate

determination of unit costs, the costs will be based on data provided by two hospital administrations (one university centre and one general hospital) to account for the actual input of personnel, material and overhead over hospital resources used. Cost calculations will be used to determine cost of interventions (surgical, endoscopic or radiological) and diagnostic imaging. The cost analysis will be reported separately from the main study manuscript.

### Patient and public involvement

The patient advocacy organisation *'Alvleeskliervereniging Nederland'* was involved in the design of the PICUS Study. The experience of the patient advocacy organisation with IAP and participation in scientific research has driven the research question and design of the study with regards to patient burden. The patient advocacy organisation will also be involved in the dissemination and implementation of the study results.

All patients eligible for participation will be asked to give written informed consent.

## ETHICS AND DISSEMINATION

The PICUS Study is conducted according to the principles of the Declaration of Helsinki (October 2013) and to the Guideline for Good Clinical Practice by the International Council for Harmonization (9 November 2016).

The need for ethical approval was waived by the Medical Ethics Review Committee of the Academic Medical Center on 28 May 2018 (W18_161 # 18.199), by the Medical Research Ethics Committee of the University Medical Center Utrecht on 4 July 2018 (18-469), by the Research Ethics Committee of Radboud University Medical Center on 23 July 2018 (2018-4520), by the Medical Ethics Review Committee of the Erasmus Medical Center on 30 July 2018 (MEC-2018-1293) and by the Medical Ethics Review Committee of the Maastricht University Medical Center on 7 September 2018 (2018-0685). Before start of inclusion, local board approval will be obtained in all participating centres.

The results of the PICUS Study will be submitted for publication in an international peer-reviewed scientific journal, regardless of study outcomes.

## DISCUSSION

Previous research has suggested that EUS might be beneficial in the detection of an aetiology in presumed IAP. However, data lack on the efficacy of routine EUS in patients with a first episode of presumed IAP, after repeat imaging after clinical recovery is negative for an aetiology. The PICUS Study aims to determine whether routine EUS is warranted in a first episode of acute pancreatitis where no cause could be uncovered after complete standard diagnostic work-up.

Currently, guidelines do not clearly define criteria for biliary origin.[11] However, it is generally agreed on that cholelithiasis, microlithiasis or biliary sludge constitutes biliary aetiology. Several previous studies have shown an association between elevated ALT levels and acute biliary pancreatitis,[32–35] with a positive predictive value of 85% for an ALT >150 U/L within 48 hours after the onset of symptoms.[11 32 33 35] Therefore, an elevated blood serum ALT level at admission is considered to entail a high probability of biliary aetiology, and pancreatitis with an elevated ALT is treated as being of biliary origin.[32–34 36] However, the majority of current literature on EUS did not exclude patients based on ALT level at admission.[15 25 26 32 37–46] As these patients have a higher a priori chance of confirmation of biliary aetiology on EUS, the aetiology detection rate of EUS might be overestimated in these studies. In PICUS, biliary aetiology is defined as either the signs of cholelithiasis, microlithiasis or biliary sludge on transabdominal ultrasonography or transient elevation of the blood serum ALT level of more than twice the upper limit of normal at admission in the absence of ALT elevating comorbidity. By only including patients with normal or slightly elevated ALT levels at admission, the aetiology detection rate as reported in PICUS will reflect the detection rate in patients who are truly considered as having presumed IAP after standard diagnostic work-up.

Multiple definitions for IAP are maintained in literature.[47] For PICUS, the definition provided by the IAP/APA evidence-based guidelines on management of acute pancreatitis was used.[11] These guidelines advise a repeat transabdominal ultrasound after clinical recovery in the work-up of presumed IAP because the index transabdominal ultrasound is less sensitive during the acute phase of pancreatitis. The subpar visualisation of gall bladder, bile ducts and pancreas is often due to excessive amounts of air in the intestines caused by pancreatitis-induced ileus and/or suboptimal cooperation of painful patients.[48] After the first episode of acute pancreatitis, repeating a transabdominal ultrasound may be able to detect biliary stones where it could not during index admission.[49] Of the current literature on EUS in IAP, however, only a minority of studies included repeat imaging in the diagnostic work-up before EUS.[15 40 41 43] Previous research has shown that a repeat transabdominal ultrasound has a diagnostic yield of 20% in patients with a first episode of IAP.[13] Omitting repeat imaging from diagnostic work-up before EUS may lead to an overestimation of the diagnostic yield of EUS. In PICUS, all patients are required to undergo imaging after clinical recovery, that is, transabdominal ultrasound or MRI/MRCP. CT is not considered sufficient imaging as biliary disease, the most common underlying aetiology in presumed IAP, cannot always be adequately detected using CT.

It is well documented that the overall diagnostic yield of EUS in patients with recurrent pancreatitis is superior to the diagnostic yield of both s-MRCP and non-s-MRCP.[18 44 46 50] In the subgroup of patients with a pancreas divisum, however, s-MRCP is considered to be superior in diagnostic yield to both EUS and MRCP.[18] The role of pancreas divisum in the aetiology of pancreatitis is unclear. Epidemiological studies have shown that the prevalence

of pancreas divisum in the general population is equal to the prevalence in patients with presumed IAP.[23] In patients with a pancreas divisum and acute pancreatitis, potentially other disease-modifying factors add to the occurrence of pancreatitis, such as increased sensitivity to toxins or genetic susceptibility. Because of this ambiguity, pancreas divisum in patients with a first episode of acute pancreatitis is mostly left untreated in clinical practice. However, if patients with a pancreas divisum present with multiple episodes of presumed IAP, the divisum is often considered to be related to the pancreatitis and is subsequently treated, often with ERCP with endoscopic sphincterotomy, although evidence supporting this practice is limited.[23] Because of both the diagnostic superiority of EUS in recurrent pancreatitis as well as the lack of clinical consequences of s-MRCP in patients with a first episode of pancreatitis, EUS is preferred to s-MRCP as the first choice for additional diagnostic testing for aetiology in patients with presumed IAP.[18 44 46 50] Subsequently, current guidelines advise performing MRCP in case of recurrent IAP after EUS fails to determine an aetiology.[11] Therefore, in PICUS, we have chosen not to systematically include s-MRCP in the diagnostic work-up before EUS of first episode of IAP.

Current guidelines advise consideration of EUS after a first or second attack of IAP.[11] However, there is a paucity of evidence on the efficacy of EUS in first episode of IAP. Three previous studies prospectively reported on EUS in patients with first episode of IAP.[25 26 38] However, in these studies, patients were not excluded based on liver enzyme abnormalities suggestive of biliary disease and no repeat imaging after clinical recovery was performed. PICUS will be the first prospective cohort study in which EUS will be performed in patients with a first episode of IAP after complete standard diagnostic work-up before EUS according to current guidelines.[11]

A diagnostic yield of 10% for any aetiology will be considered reasonable to justify incorporating routine EUS after a first episode of presumed IAP. This cut-off value was determined during a multidisciplinary meeting of the Dutch Pancreatitis Study Group, which included the principal investigators of several trials being executed by the Dutch Pancreatitis Study Group. Considering the expectation that the majority of uncovered aetiologies by EUS will be treatable (eg, biliary disease) and adequate treatment could prevent pancreatitis recurrence, while in a minority of uncovered aetiologies diagnosis before progression of disease might be crucial for prognosis (eg, malignancy), a positive result in 10% of patients was deemed sufficient to warrant routine EUS after a first episode of presumed IAP.

In conclusion, the PICUS Study is the first prospective cohort study of patients with a single episode of presumed IAP after complete standard diagnostic work-up (including exclusion based on blood serum ALT and imaging after clinical recovery). The results of the PICUS study will establish whether routine EUS should be incorporated in the guidelines for standard diagnostic work-up after a first episode of presumed IAP.

**Author affiliations**
[1]Department of Gastroenterology and Hepatology, Amsterdam University Medical Centres, Amsterdam, The Netherlands
[2]Research and Development, Saint Antonius Hospital, Nieuwegein, Utrecht, The Netherlands
[3]Department of Surgery, Saint Antonius Hospital, Nieuwegein, Utrecht, The Netherlands
[4]Department of Gastroenterology and Hepatology, Erasmus Medical Center, Rotterdam, Zuid-Holland, The Netherlands
[5]Department of Surgery, Maastricht UMC+, Maastricht, Limburg, The Netherlands
[6]Department of Gastroenterology and Hepatology, Franciscus Gasthuis en Vlietland, Rotterdam, Zuid-Holland, The Netherlands
[7]Department of Gastroenterology and Hepatology, HagaZiekenhuis, Den Haag, Zuid-Holland, The Netherlands
[8]Department of Gastroenterology and Hepatology, Martini Ziekenhuis, Groningen, Groningen, The Netherlands
[9]Department of Surgery, Amsterdam University Medical Centres, Amsterdam, Noord-Holland, The Netherlands
[10]Department of Gastroenterology and Hepatology, Meander MC, Amersfoort, Utrecht, The Netherlands
[11]Department of Gastroenterology and Hepatology, Maasstad Hospital, Rotterdam, Zuid-Holland, The Netherlands
[12]Department of Gastroenterology and Hepatology, Catharina Hospital, Eindhoven, North Brabant, The Netherlands
[13]Department of Gastroenterology and Hepatology, UMCG, Groningen, Groningen, The Netherlands
[14]Department of Gastroenterology and Hepatology, Spaarne Gasthuis, Haarlem, Noord-Holland, The Netherlands
[15]Department of Gastroenterology and Hepatology, Gelre Ziekenhuizen, Apeldoorn, Gelderland, The Netherlands
[16]Department of Gastroenterology and Hepatology, Radboud university medical center, Nijmegen, the Netherlands
[17]Department of Gastroenterology and Hepatology, Elisabeth-TweeSteden Ziekenhuis, Tilburg, Noord-Brabant, The Netherlands
[18]Department of Gastroenterology and Hepatology, Maastricht UMC+, Maastricht, Limburg, The Netherlands
[19]Department of Gastroenterology and Hepatology, LUMC, Leiden, Zuid-Holland, The Netherlands
[20]Department of Gastroenterology and Hepatology, Noordwest Ziekenhuisgroep, Alkmaar, Noord-Holland, The Netherlands
[21]Department of Gastroenterology and Hepatology, OLVG, Amsterdam, Noord-Holland, The Netherlands
[22]Department of Gastroenterology and Hepatology, Medisch Centrum Haaglanden, Den Haag, Zuid-Holland, The Netherlands
[23]Department of Gastroenterology and Hepatology, Reinier de Graaf Groep, Delft, Zuid-Holland, The Netherlands
[24]Department of Gastroenteroloy and Hepatology, Jeroen Bosch Hospital, 's-Hertogenbosch, Noord-Brabant, The Netherlands
[25]Department of Surgery, University Medical Center Utrecht, Utrecht, the Netherlands
[26]Department of Gastroenterology and Hepatology, Canisius Wilhelmina Hospital, Nijmegen, Gelderland, The Netherlands
[27]Department of Gastroenterology and Hepatology, Albert Schweitzer Ziekenhuis, Dordrecht, Zuid-Holland, The Netherlands
[28]Department of Gastroenterology and Hepatology, Medisch Spectrum Twente, Enschede, Overijssel, The Netherlands
[29]Department of Gastroenterology and Hepatology, University Medical Center Utrecht, Utrecht, the Netherlands
[30]Department of Gastroenterology and Hepatology, Diakonessenhuis Utrecht Zeist Doorn, Utrecht, Utrecht, The Netherlands
[31]Department of Gastroenterology and Hepatology, Ziekenhuis Gelderse Vallei, Ede, Gelderland, The Netherlands
[32]Department of Gastroenterology and Hepatology, Saint Antonius Hospital, Nieuwegein, Utrecht, The Netherlands
[33]AMC, Amsterdam, North Holland, The Netherlands

**Acknowledgements** The authors would like to acknowledge: all of the members of the Dutch Pancreatitis Study Group, for their continuous efforts in recruitment of patients, Dr S. van Dieren, for providing her expertise in the calculation of the sample size, Dr M. van der Vlugt, for her willingness to act as independent physician for the PICUS Study, and the pancreatic disease patient advocacy organisation *Alvleeskliervereniging Nederland*, for their effort in representing the perspective of (participating) patients during the study design.

**Contributors** DSU drafted the manuscript. HCT, RCV, SAB, MGB and JvH co-authored the writing of the manuscript. DSU, RCV, SAB, MAB, MB, PF, EJMvG, J-WP, HCvS, FV, MGB and JvH were involved in the design of the study during several meetings of the Dutch Pancreatitis Study Group. NDH, M-PGA, AB, RAB, MB, LH, WLC, HMvD, BCvE, GWE, WLH, CVH, AI, LMK, SDK, LEP, RQ, TER, ACT, AYT, NV, AMV, RLvW and BJW critically assessed the study design, during several meetings, and edited the manuscript. All authors read and approved the final manuscript.

**Funding** This work was supported by the Dutch Digestive Disease Foundation (*Maag Lever Darm Stichting*, grant number D17-25). The PICUS Study is an investigator-initiated study.

**Competing interests** None declared.

**Patient consent for publication** Not required.

**Provenance and peer review** Not commissioned; externally peer reviewed.

**ORCID iD**
Devica S Umans http://orcid.org/0000-0002-4872-8389

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
