## [Reviewer comments · BMJ Open]

ARTICLE DETAILS

TITLE (PROVISIONAL)	The role of endoscopic ultrasonography in the diagnostic work-up of idiopathic acute pancreatitis (PICUS): study protocol for a nationwide prospective cohort study
AUTHORS	Umans, Devica; Timmerhuis, Hester; Hallensleben, Nora; Bouwense, Stefan; Anten, Marie-Paule; Bhalla, Abha; Bijlsma, Rina; Boermeester, Marja; Brink, Menno; Hol, Lieke; Bruno, Marco; Curvers, Wouter; van Dullemen, Hendrik; van Eijck, Brechje; Erkelens, G; Fockens, Paul; van Geenen, Erwin; Hazen, Wouter; Hoge, Chantal; Inderson, Akin; Kager, Liesbeth; Kuiken, Sjoerd; Perk, Lars; Poley, Jan-Werner; Quispel, Rutger; Römkens, Tessa; van Santvoort, Hjalmar; Tan, Adriaan; Thijssen, Annemieke; Venneman, Niels; Vleggaar, Frank; Voorburg, Annet; van Wanrooij, Roy; Witteman, Ben; Verdonk, Robert; Besselink, Marc; van Hooft, Jeanin

VERSION 1 - REVIEW

REVIEWER	Bimal Chandra Shil Sir Salimullah Medical College Dhaka Bangladesh
REVIEW RETURNED	25-Nov-2019

GENERAL COMMENTS	The work is really appreciable. It will help to make the guideline for managing Idiopathic Acute Pancreatitis(IAP). However minimal correction is needed which is mentioned in attached file. During referencing , If there are more than 6 authors, list the first 6 authors followed by "et al." The reviewer provided a marked copy with additional comments. Please contact the publisher fo full details.
---

REVIEWER	Piyush Somani Department of Gastroenterology, Thumbay Hospital, Dubai.
REVIEW RETURNED	22-Dec-2019

GENERAL COMMENTS	It was a pleasure reviewing this study protocol. This study protocol will investigate the role of EUS after first episode of idiopathic acute
---

pancreatitis (IAP).

There are few comments to make.

1) What will be defined as clinical recovery from acute pancreatitis?

2) There are multiple definitions of IAP in literature which is not standardized.

Please go through the article "Somani P, Sunkara T, Sharma M. Role of endoscopic ultrasound in idiopathic pancreatitis. *World J Gastroenterol* 2017 Oct 14;23(38):6952-6961".

Please clarify the definition used in the present study protocol.

CT abdomen or MRCP is included in most of the definitions of IAP.

However, in the 'BACKGROUND' section of study protocol, the definition includes only transabdominal ultrasound (Bakker OJ, van Brunschot S, van Santvoort HC, Besselink MG, Bollen TL, Boermeester MA, et al. Early versus on-demand nasoenteric tube feeding in acute pancreatitis. *N Engl J Med* 2014;371(21):1983-93).

3) Regarding the 20 percent diagnostic yield of repeat ultrasound as mentioned in the 'BACKGROUND' section of study protocol, (Hallensleben ND, Umans DS, Bouwense S, Verdonk RC, Besselink MG, Van Hooft JE, et al. The clinical course and diagnostic work-up of idiopathic acute pancreatitis, a post-hoc analysis of a prospective multicenter observational cohort. *Digestive Disease Week; San Diego: Gastroenterology*) there are no other studies to support that. Please comment on that.

4) In the 'FOLLOW UP' section of study protocol, secretin based MRCP will be performed if there is repeat acute pancreatitis and EUS is negative for etiology to look for pancreas divisum. However, there are studies to show that EUS is quite sensitive/specific for detection of pancreas divisum (Kushnir VM, Wani SB, Fowler K, Menias C, Varma R,

Narra V, et al. Sensitivity of endoscopic ultrasound, multidetector computed tomography, and magnetic resonance cholangiopancreatography in the diagnosis of pancreas divisum: a tertiary center experience. *Pancreas*. 2013;42(3):436-41.

Sharma M, Pathak A, Rameshbabu CS, Rai P, Kirnake V, Shoukat A. Imaging of pancreas divisum by linear-array endoscopic ultrasonography. *Endosc Ultrasound* 2016; 5: 21-29.

Rana SS, Bhasin DK, Sharma V, Rao C, Singh K. Role of endoscopic ultrasound in the diagnosis of pancreas divisum. *Endosc Ultrasound* 2013; 2: 7-10) and also at the same time there is lots of controversy regarding the role of pancreas divisum in the etiology of recurrent acute pancreatitis where many of recent papers has questioned its role.

(DiMagno MJ, Wamsteker E-J. Pancreas divisum. *Curr Gastroenterol Rep*. 2011;13(2):150-6.

Bertin C, Pelletier A-L, Vullierme MP, Bienvenu T, Rebours V, Hentic O, et al. Pancreas divisum is not a cause of pancreatitis by itself but acts as a partner of genetic mutations. *Am J Gastroenterol*. 2012;107(2):311-7.

Fogel EL, Toth TG, Lehman GA, DiMagno MJ, DiMagno EP. Does endoscopic therapy favorably affect the outcome of patients who have recurrent acute pancreatitis and pancreas divisum? *Pancreas*. 2007;34(1):21-45.

Liao Z, Gao R, Wang W, Ye Z, Lai X-W, Wang X-T, et al. A systematic review on endoscopic detection rate, endotherapy, and surgery for pancreas divisum. *Endoscopy*. 2009;41(5):439-44.

Chacko LN, Chen YK, Shah RJ. Clinical outcomes and nonendoscopic interventions after minor papilla

	endotherapy in patients with symptomatic pancreas divisum. Gastrointest Endosc. 2008;68(4):667–73). So the query: Is there need of secretin stimulated MRCP once EUS is negative just to look for pancreas divisum when EUS is good to detect the same with its role being controversial? 5) 'In PICUS, biliary etiology is defined as either the signs of cholelithiasis, microlithiasis or biliary sludge on transabdominal ultrasonography, or transient elevation of the blood serum ALT level of more than twice the upper limit of normal at admission'. However, not all patients with twice elevated ALT are having biliary etiology for acute pancreatitis. As per studies quoted in the 'Discussion' section, positive predictive value is 85% for an ALT > 150 U/L within 48 hours after onset of symptoms. This means still 15 percent can have other etiology. This requires further clarification by the authors. 6) 'In PICUS, all patients are required to undergo imaging after clinical recovery, i.e. transabdominal ultrasound or MRI/MRCP'. However, biliary sludge can form after clinical recovery after acute pancreatitis or due to use of antibiotics like ceftriaxone. This may overestimate the biliary etiology in the present study protocol.
--	---

REVIEWER	Paolo Aseni ASST Grande Ospedale Metropolitano, Niguarda Hospital, Milan, Italy
REVIEW RETURNED	15-Jan-2020

GENERAL COMMENTS	Dear Editor, thank you for inviting me to review this paper entitled "The role of endoscopic ultrasonography in the diagnostic work-up of idiopathic acute pancreatitis (PICUS): study protocol for a nationwide prospective cohort study". The article is a very nice study protocol to investigate the diagnostic yield of endoscopic ultrasonography (EUS) to detect a cause of pancreatitis in patients with presumed idiopathic acute pancreatitis (IAP) after the first diagnostic work-up. The protocol seems quite well organized according to SPIRIT guidelines. This prospective study is justified on the assumption that the aetiology of acute pancreatitis may have a crucial impact on treatment policy, and it should be determined promptly and accurately. I have some concern about the undefined time framing within which the first diagnostic path should be completed. In patients with microlithiasis, the progression of biliary sludge and small stones into the common bile duct and then in the duodenum can be, at least in part, time-dependent. For this reason, a potential first bias in the identification of potential IAP among different Centres can occur. It should be better to have a specific time framing for all participating Centres specifically for US and RMI. Another critical concern is the definition of "idiopathic acute pancreatitis" in the absence of a specific diagnostic algorithm for all Centres participating in the study. A definite diagnostic algorithm should be helpful concerning some possible, less frequent aetiologies of acute pancreatitis. A variable percentage (from 10% to 30%) of patients with acute pancreatitis does not have an established etiologic factor after routine investigation, and these patients are classified as having idiopathic acute pancreatitis (IAP).
---

	Extensive evaluations, including a uniform algorithm with biochemical tests, EUS and MRCP, can be a useful unified diagnostic strategy. In patients with suspected IAP some Authors recommend intravenous secretin combined with MRCP before EUS for better visualization of the pancreatic duct morphology, which can also provide indirect functional information about the Oddi's sphincter dysfunction. I had the impression that in this study protocol, a diagnostic work-up is only suggested and not explicitly and uniformly organized and structured for all participating Centres. Some rare forms of pancreatitis are not considered in the work-up (vasculitis and rheumatic diseases associated pancreatitis, viral infections such as EBV, Hepatitis E, CMV). Is there a specific reason for that? My last concern is the lack of an accurate description of the statistical methodology about "the intention to treat analysis" and "per protocol analysis"; this can have relevance in the case that the drop-out is much higher than 10%.
--	---

VERSION 1 – AUTHOR RESPONSE

REVIEWERS(S)' COMMENTS TO AUTHOR:

Reviewer: 1

Reviewer Name: Bimal Chandra Shil

Institution and Country:

Sir Salimullah Medical College

Dhaka

Bangladesh

Please state any competing interests or state 'None declared': None Declared

Please leave your comments for the authors below

The work is really appreciable. It will help to make the guideline for managing Idiopathic Acute Pancreatitis(IAP). However minimal correction is needed which is mentioned in attached file.

During referencing , If there are more than 6 authors, list the first 6 authors followed by "et al."

Authors: We would like to thank dr. Shil for his kind comments. We have redrafted the manuscript in accordance with dr. Shil's suggestions, we have corrected the reference lists and we have included the questionnaire in an additional file.

Reviewer: 2

Reviewer Name: Piyush Somani

Institution and Country:

Department of Gastroenterology,

Thumbay Hospital, Dubai.

Please state any competing interests or state 'None declared': None declared

Please leave your comments for the authors below

It was a pleasure reviewing this study protocol. This study protocol will investigate the role of EUS after first episode of idiopathic acute pancreatitis (IAP).

There are few comments to make.

1) What will be defined as clinical recovery from acute pancreatitis?

Authors: We would like to thank Dr. Somani for his elaborate assessment of our study protocol. Clinical recovery, in this study, is defined as resolution of pancreatic inflammation and is present when one of the following criteria is met:

1. Discharge from the hospital
2. Normal inflammation parameters in laboratory tests
3. No signs of pancreatic inflammation on imaging

We have included this definition in the revised version of additional file 2 (“Relevant Definitions”).

2) There are multiple definitions of IAP in literature which is not standardized. Please go through the article “Somani P, Sunkara T, Sharma M. Role of endoscopic ultrasound in idiopathic pancreatitis. *World J Gastroenterol* 2017 Oct 14;23(38):6952-6961”. Please clarify the definition used in the present study protocol. CT abdomen or MRCP is included in most of the definitions of IAP. However, in the ‘BACKGROUND’ section of study protocol, the definition includes only transabdominal ultrasound (Bakker OJ, van Brunschot S, van Santvoort HC, Besselink MG, Bollen TL, Boermeester MA, et al. Early versus on-demand nasoenteric tube feeding in acute pancreatitis. *N Engl J Med* 2014;371(21):1983-93).

Authors: We agree with dr. Somani’s statement that there is much ambiguity, both in research as well as in clinical practice, regarding the definition of IAP. For this study, we have chosen to use the definition maintained by the IAP/APA Guidelines (reference 11 in the manuscript). The IAP/APA evidence-based guidelines on management of acute pancreatitis state that acute pancreatitis can be classified as of idiopathic origin when no etiology is found in standard diagnostic work-up, which comprises a detailed personal/family history, laboratory tests including blood serum liver enzymes, triglycerides and calcium levels, and imaging including transabdominal ultrasonography at admission and transabdominal ultrasonography after clinical recovery. We have clarified this in the “Eligibility criteria” section of the revised manuscript and in additional file 2 (“Relevant Definitions”). The reasons to consider MRI, MRCP and CT within our study protocol is extensively discussed in the Discussion section and below under comment 4.

3) Regarding the 20 percent diagnostic yield of repeat ultrasound as mentioned in the ‘BACKGROUND’ section of study protocol, (Hallensleben ND, Umans DS, Bouwense S, Verdonk RC, Besselink MG, Van Hooft JE, et al. The clinical course and diagnostic work-up of idiopathic acute pancreatitis, a post-hoc analysis of a prospective multicenter observational cohort. *Digestive Disease Week; San Diego: Gastroenterology*) there are no other studies to support that. Please comment on that.

Authors: Dr. Somani is correct to state that there is a paucity in literature regarding the diagnostic yield of a second transabdominal ultrasound in patients with IAP. However, there is data from the Netherlands indicating that the repeat ultrasound has a diagnostic yield of 20%. This article is accepted by the *United European Gastroenterology Journal* and is currently in press. This reference is now corrected.

4) In the ‘FOLLOW UP’ section of study protocol, secretin based MRCP will be performed if there is repeat acute pancreatitis and EUS is negative for etiology to look for pancreas divisum. However, there are studies to show that EUS is quite sensitive/specific for detection of pancreas divisum (Kushnir VM, Wani SB, Fowler K, Menias C, Varma R, Narra V, et al. Sensitivity of endoscopic ultrasound, multidetector computed tomography, and magnetic resonance cholangiopancreatography in the diagnosis of pancreas divisum: a tertiary center experience. *Pancreas*. 2013;42(3):436–41. Sharma M, Pathak A, Rameshbabu CS, Rai P, Kirnake V, Shoukat A. Imaging of pancreas divisum by linear-array endoscopic ultrasonography. *Endosc Ultrasound* 2016; 5: 21–29. Rana SS, Bhasin DK, Sharma V, Rao C, Singh K. Role of endoscopic ultrasound in the diagnosis of pancreas divisum. *Endosc Ultrasound* 2013; 2: 7–10) and also at the same time there is

lots of controversy regarding the role of pancreas divisum in the etiology of recurrent acute pancreatitis where many of recent papers has questioned its role. (DiMagno MJ, Wamsteker E-J. Pancreas divisum. *Curr Gastroenterol Rep.* 2011;13(2):150–6. Bertin C, Pelletier A-L, Vullierme MP, Bienvenu T, Rebours V, Hentic O, et al. Pancreas divisum is not a cause of pancreatitis by itself but acts as a partner of genetic mutations. *Am J Gastroenterol.* 2012;107(2):311–7. Fogel EL, Toth TG, Lehman GA, DiMagno MJ, DiMagno EP. Does endoscopic therapy favorably affect the outcome of patients who have recurrent acute pancreatitis and pancreas divisum? *Pancreas.* 2007;34(1):21–45. Liao Z, Gao R, Wang W, Ye Z, Lai X-W, Wang X-T, et al. A systematic review on endoscopic detection rate, endotherapy, and surgery for pancreas divisum. *Endoscopy.* 2009;41(5):439–44. Chacko LN, Chen YK, Shah RJ. Clinical outcomes and nonendoscopic interventions after minor papilla endotherapy in patients with symptomatic pancreas divisum. *Gastrointest Endosc.* 2008;68(4):667–73). So the query: Is there need of secretin stimulated MRCP once EUS is negative just to look for pancreas divisum when EUS is good to detect the same with its role being controversial?

Authors: We agree with dr. Somani that the role of pancreas divisum is highly controversial, and we have elaborated on this topic in the Discussion section. As we have chosen to follow the recommendations of the IAP/APA guidelines, we have included the IAP/APA Guideline advice to perform (secretin-enhanced) MRCP as a second step to identify morphologic abnormalities, but only in recurrent IAP (as the evidence for pancreas divisum as an etiological factor in first episode IAP is even more scarce than in recurrent IAP). However, the question whether (secretin-enhanced) MRCP has an adequate diagnostic yield after negative EUS, or whether the potential findings by MRCP are clinically relevant, are beyond the scope of this study, which is powered on the diagnostic yield of EUS. We would be pleased to examine these questions in future studies, and data on use and yield of MRCP in patients participating in the PICUS study may be of use in the design of such future research projects.

5) 'In PICUS, biliary etiology is defined as either the signs of cholelithiasis, microlithiasis or biliary sludge on transabdominal ultrasonography, or transient elevation of the blood serum ALT level of more than twice the upper limit of normal at admission'. However, not all patients with twice elevated ALT are having biliary etiology for acute pancreatitis. As per studies quoted in the 'Discussion' section, positive predictive value is 85% for an ALT > 150 U/L within 48 hours after onset of symptoms. This means still 15 percent can have other etiology. This requires further clarification by the authors.

Authors: We would like to thank dr. Somani for pointing out this important error. We agree that comorbidity (e.g. hepatitis) could alter ALT levels, and the ALT levels may subsequently be wrongly interpreted as indicative of biliary pancreatitis. It is certainly not our intention to exclude these patients on the basis of having biliary pancreatitis. Therefore, in PICUS, we will only consider elevated ALT levels as indicative of biliary pancreatitis when they are transiently elevated at admission and normalize during admission, as is typical for biliary pancreatitis, and when there is no ALT altering comorbidity. We have clarified this matter in additional file 2 ("Relevant Definitions") and in the Discussion section.

6) 'In PICUS, all patients are required to undergo imaging after clinical recovery, i.e. transabdominal ultrasound or MRI/MRCP'. However, biliary sludge can form after clinical recovery after acute pancreatitis or due to use of antibiotics like ceftriaxone. This may overestimate the biliary etiology in the present study protocol.

Authors: This is an interesting perspective. It is true that biliary sludge could form after recovery from the acute pancreatitis, and EUS may then wrongly suggest that the pancreatitis was of biliary origin. This is important, because acute biliary pancreatitis is a clear indication for cholecystectomy, while it

is widely believed asymptomatic cholelithiasis or biliary sludge is not an indication for treatment. Thus, overestimation of biliary etiology by EUS may lead to unnecessary cholecystectomies. At the same time, it could be hypothesized that IAP may be caused by a solitary stone or a minimal amount of microlithiasis or sludge, which may have passed to the duodenum by the time of the EUS, leading to a false-negative result of EUS and subsequently an underestimation of biliary etiology. By following patients for a year after EUS, we can assess whether cholecystectomy after an EUS positive for biliary etiology reduces pancreatitis recurrence rate, as is the case in biliary pancreatitis treated by cholecystectomy in comparison with biliary pancreatitis not treated by cholecystectomy. Although this study is not powered on this endpoint, this may give insight in the extent of overestimation of biliary etiology when performing EUS after clinical recovery, and may be useful in the design of future studies tackling the effect of timing of EUS on diagnostic yield of clinically relevant abnormalities.

Reviewer: 3

Reviewer Name: Paolo Aseni

Institution and Country: ASST Grande Ospedale Metropolitano, Niguarda Hospital, Milan, Italy

Please state any competing interests or state 'None declared': None declared

Please leave your comments for the authors below

Dear Editor, thank you for inviting me to review this paper entitled "The role of endoscopic ultrasonography in the diagnostic work-up of idiopathic acute pancreatitis (PICUS): study protocol for a nationwide prospective cohort study". The article is a very nice study protocol to investigate the diagnostic yield of endoscopic ultrasonography (EUS) to detect a cause of pancreatitis in patients with presumed idiopathic acute pancreatitis (IAP) after the first diagnostic work-up.

The protocol seems quite well organized according to SPIRIT guidelines.

This prospective study is justified on the assumption that the aetiology of acute pancreatitis may have a crucial impact on treatment policy, and it should be determined promptly and accurately.

I have some concern about the undefined time framing within which the first diagnostic path should be completed. In patients with microlithiasis, the progression of biliary sludge and small stones into the common bile duct and then in the duodenum can be, at least in part, time-dependent. For this reason, a potential first bias in the identification of potential IAP among different Centres can occur. It should be better to have a specific time framing for all participating Centres specifically for US and RMI.

Authors: We would like to thank dr. Aseni for his considerate feedback and this interesting point. We agree with dr. Aseni, and dr. Somani, who raises a similar issue in his sixth comment, that timing of EUS may influence diagnostic yield, as we have elaborated on in our response to dr. Somani above. We have deliberated whether the ultrasound or MRI/MRCP should be performed within a set time frame. Our reason not to do so is twofold. First, as acute pancreatitis is known to have a varying clinical course, with some patients only admitted for a few days, while others may have a hospital stay of multiple months, it is difficult to set a specific time frame which would have been suitable for all patients. Excluding patients in whom it was not possible to perform complete diagnostic work-up within the time frame, either due to severe pancreatitis or due to comorbidity, may lead to selection bias. Second, it was our specific aim to determine the diagnostic yield of EUS in routine practice. In clinical practice, not only due to patient or disease related factors but potentially also due to waiting periods, it may not be possible to perform imaging within a certain time frame. In order to determine the true diagnostic yield of EUS in a routine setting, we chose the approach which most closely mimicked this routine setting.

Another critical concern is the definition of "idiopathic acute pancreatitis" in the absence of a specific diagnostic algorithm for all Centres participating in the study. A definite diagnostic algorithm should be helpful concerning some possible, less frequent aetiologies of acute pancreatitis. A variable percentage (from 10% to 30%) of patients with acute pancreatitis does not have an established

etiologic factor after routine investigation, and these patients are classified as having idiopathic acute pancreatitis (IAP). Extensive evaluations, including a uniform algorithm with biochemical tests, EUS and MRCP, can be a useful unified diagnostic strategy. In patients with suspected IAP some Authors recommend intravenous secretin combined with MRCP before EUS for better visualization of the pancreatic duct morphology, which can also provide indirect functional information about the Oddi's sphincter dysfunction.

I had the impression that in this study protocol, a diagnostic work-up is only suggested and not explicitly and uniformly organized and structured for all participating Centres.

Authors: Dr. Aseni raises an important point, and we agree that it is crucial for the homogeneity of the patient cohort in this study that the diagnostic work-up is uniform. This is also the case in our study, but we have not stated this clearly enough in our protocol. We have amended our "Study population" section in the hope this clarifies the fact that the diagnostic tests as laid out in table 1 are to be performed in all subjects and these tests cannot show any signs of an etiology in all subjects.

Some rare forms of pancreatitis are not considered in the work-up (vasculitis and rheumatic diseases associated pancreatitis, viral infections such as EBV, Hepatitis E, CMV). Is there a specific reason for that?

Authors: This is an interesting query. We have debated to what extent uncommon etiologies should be excluded before performing EUS. In this study, we aimed to determine whether EUS should be a routine part of standard diagnostic work-up in first episode IAP. Standard diagnostic work-up, as defined by the IAP/APA evidence-based guideline on management of acute pancreatitis (reference 11 of the manuscript), does not include screening for rare etiologies such as the ones you mentioned, and we suspect it is currently not clinical practice to rule out these rare etiologies before performing EUS. We want to determine what the true value is of EUS in clinical practice. Extensive screening, imaging and/or genetic counselling before EUS may lead to exclusion of patients who would have undergone EUS when following the IAP/APA guidelines. As EUS would have been negative in these patients, exclusion of these patients could lead to an overestimation of the diagnostic yield of EUS, which may have been the case in some of the existing literature. Thus, in order to determine the true diagnostic yield in this patient group, we have chosen to follow the IAP/APA guidelines.

My last concern is the lack of an accurate description of the statistical methodology about "the intention to treat analysis" and "per protocol analysis"; this can have relevance in the case that the drop-out is much higher than 10%.

Authors: We would like to thank dr. Aseni for pointing out this ambiguity. In the "Statistical aspects" section of the manuscript, we have clarified how we have defined the per-protocol population (i.e. meeting all eligibility criteria including all of the diagnostic tests required for the diagnosis of IAP, undergoing EUS as described in the "Endoscopic ultrasonography section").

VERSION 2 – REVIEW

REVIEWER	piyush somani Department of gastroenterology, Thumbay Hospital, Dubai, UAE.
REVIEW RETURNED	22-Mar-2020

GENERAL COMMENTS	1) There are multiple definitions of IAP in literature which is not standardized. Please go through the article "Somani P, Sunkara T, Sharma M.
--

	Role of endoscopic ultrasound in idiopathic pancreatitis. World J Gastroenterol 2017 Oct 14;23(38):6952-6961". Please clarify the definition used in the present study protocol. CT abdomen or MRCP is included in most of the definitions of IAP. However, in the 'BACKGROUND' section of study protocol, the definition includes only transabdominal ultrasound (Bakker OJ, van Brunschot S, van Santvoort HC, Besselink MG, Bollen TL, Boermeester MA, et al. Early versus on-demand nasoenteric tube feeding in acute pancreatitis. N Engl J Med 2014;371(21):1983-93). 2) Regarding the 20 percent diagnostic yield of repeat ultrasound as mentioned in the 'BACKGROUND' section of study protocol, (Hallensleben ND, Umans DS, Bouwense S, Verdonk RC, Besselink MG, Van Hooft JE, et al. The clinical course and diagnostic work-up of idiopathic acute pancreatitis, a post-hoc analysis of a prospective multicenter observational cohort. Digestive Disease Week; San Diego: Gastroenterology) there are no other studies to support that. Please comment on that. 3) 'In PICUS, biliary etiology is defined as either the signs of cholelithiasis, microlithiasis or biliary sludge on transabdominal ultrasonography, or transient elevation of the blood serum ALT level of more than twice the upper limit of normal at admission'. However, not all patients with twice elevated ALT are having biliary etiology for acute pancreatitis. As per studies quoted in the 'Discussion' section, positive predictive value is 85% for an ALT > 150 U/L within 48 hours after onset of symptoms. This means still 15 percent can have other etiology. This requires further clarification by the authors. 4) 'In PICUS, all patients are required to undergo imaging after clinical recovery, i.e. transabdominal ultrasound or MRI/MRCP'. However, biliary sludge can form after clinical recovery after acute pancreatitis or due to use of antibiotics like ceftriaxone. This may overestimate the biliary etiology in the present study protocol.
--	--

REVIEWER	Paolo Aseni Ospedale Niguarda
REVIEW RETURNED	05-Mar-2020

GENERAL COMMENTS	Dear Editor, thank you for inviting me to review again this interesting study protocol entitled "The role of endoscopic ultrasonography in the diagnostic work-up of idiopathic acute pancreatitis (PICUS): study protocol for a nationwide prospective cohort study". The Authors have responded satisfactorily to some of the reviewers' queries. However, it should be noted that they did not answer systematically point by point to all the various aspects of criticism highlighted by reviewers. The time framing from hospitalization to EUS is not specified, and biliary sludge may produce after the acute pancreatic episode and overestimate the incidence. Furthermore, an essential aspect concerning a uniform diagnostic algorithm for all centres is not clarified: a diagnostic work-up uniformly organized and structured for all participating Centers could leave fewer biases about some rare
--

	etiologies. It remains statistically unanswered the question of whether the percentage of drop out should exceed 10%. However, with all these limitations, I really hope that this study can add some valuable information about the diagnostic yield of EUS in IAP.
--	--

VERSION 2 – AUTHOR RESPONSE

REVIEWER(S)' COMMENTS TO AUTHOR:

Reviewer: 3

Reviewer Name: Paolo Aseni

Institution and Country: Ospedale Niguarda

Please state any competing interests or state 'None declared': None declared

Please leave your comments for the authors below

Dear Editor, thank you for inviting me to review again this interesting study protocol entitled "The role of endoscopic ultrasonography in the diagnostic work-up of idiopathic acute pancreatitis (PICUS): study protocol for a nationwide prospective cohort study".

The Authors have responded satisfactorily to some of the reviewers' queries. However, it should be noted that they did not answered systematically point by point to all the various aspects of criticism highlighted by reviewers.

The time framing from hospitalization to EUS is not specified, and biliary sludge may produce after the acute pancreatic episode and overestimate the incidence. Furthermore, an essential aspect concerning a uniform diagnostic algorithm for all centres is not clarified: a diagnostic work-up uniformly organized and structured for all participating Centers could leave fewer biases about some rare etiologies. It remains statistically unanswered the question of whether the percentage of drop out should exceed 10%. However, with all these limitations, I really hope that this study can add some valuable information about the diagnostic yield of EUS in IAP.

Authors: We would like to thank dr. Aseni for his thorough review of our study protocol and his valuable feedback.

The issue he raised with regards to the time framing from hospitalization to EUS poses an interesting perspective, and is one that was also previously mentioned by dr. Aseni and Reviewer #2 in the first peer-review commentary. To our previous statements that timing could lead to both overestimation as well as underestimation of diagnostic yield, we would like to add that we have taken this limitation of the study into consideration, as is exemplified by our fifth bullet point in the "Strengths & Limitations" section: "As the timing of the EUS is set to be after clinical recovery from pancreatitis in this trial, no conclusions on the diagnostic yield of EUS in a different time frame can be drawn from this study." Additionally, we have disclosed our decision to perform EUS after clinical recovery in the Discussion section, line 375-390. Moreover, as stated in the "Baseline variables" section, we will report on the time from hospitalization to EUS. Finally, we would like to emphasize that we did not choose a set time frame from discharge to EUS, because we anticipated that this study design would not be an accurate reflection of current clinical practice as it would be logistically challenging to meet a narrow time frame. We hope to have adequately addressed this issue in these sections of the manuscript.

With regards to the uniform diagnostic algorithm for all centers: as mentioned in our response to the first peer-review, we want to emphasize that it is the case that there is an uniform diagnostic algorithm for all centers. This is detailed in the "Study population" section of the manuscript, and in table 1 and 2 and in figure 1.

With regards to a drop-out exceeding 10%, we have chosen a very conservative 10% drop-out rate for our sample size calculation. The Dutch Pancreatitis Study Group has over 15 years of research experience in acute pancreatitis patients in the Netherlands and in previous studies by the Study

Group, drop-out rates have always been lower than 5%. Thus, it seems highly unlikely the drop-out rate will exceed 10%. However, if it turns out the drop-out rate exceeds 10%, we will discuss the drop-out rate, the reasons for the unusually high drop-out rate and the effects of the drop-out rate on our results extensively in our main manuscript.

Reviewer: 2

Reviewer Name: piyush somani

Institution and Country: Department of gastroenterology, Thumbay Hospital, Dubai, UAE.

Please state any competing interests or state 'None declared': None declared

Please leave your comments for the authors below

1) There are multiple definitions of IAP in literature which is not standardized.

Please go through the article "Somani P, Sunkara T, Sharma M. Role of endoscopic ultrasound in idiopathic pancreatitis. World J Gastroenterol 2017 Oct 14;23(38):6952-6961".

Please clarify the definition used in the present study protocol.

CT abdomen or MRCP is included in most of the definitions of IAP. However, in the 'BACKGROUND' section of study protocol, the definition includes only transabdominal ultrasound (Bakker OJ, van Brunschot S, van Santvoort HC, Besselink MG, Bollen TL, Boermeester MA, et al. Early versus on-demand nasoenteric tube feeding in acute pancreatitis. N Engl J Med 2014;371(21):1983-93).

Authors: We would like to thank dr. Somani for this extensive review of our study protocol. With great interest we have read the article suggested by dr. Somani. As discussed in our first rebuttal, we agree with dr. Somani's statement that there is much ambiguity, both in research as well as in clinical practice, regarding the definition of IAP. We have addressed this in the Discussion: "Multiple definitions for IAP are maintained in literature (Somani et al., Role of endoscopic ultrasound in idiopathic pancreatitis). For PICUS, the definition provided by the IAP/APA evidence-based guidelines on management of acute pancreatitis was used."

We have clarified this in the "Eligibility criteria" section of the revised manuscript and in additional file 2 ("Relevant Definitions"). The reasons to consider MRI, MRCP and CT within our study protocol is extensively discussed in the Discussion section. Please also see the "Study Population" section of our manuscript: "The subjects of this study have had a first episode of acute pancreatitis, as defined by the 2012 Revised Atlanta criteria (20), with an unknown origin after standard diagnostic work-up, according to the 2013 International Association of Pancreatology/American Pancreatic Association (IAP/APA) evidence-based guidelines on management of acute pancreatitis (11). The diagnostic modalities that constitute standard diagnostic work-up are listed in table 1 and additional file 1. The diagnostic tests as laid out in table 1 are to be performed in all subjects and these tests cannot show any signs of an etiology in all subjects. Potential etiologies and their definitions are listed in table 2 and additional file 1." and other relevant definitions are listed in additional file 2.

2) Regarding the 20 percent diagnostic yield of repeat ultrasound as mentioned in the 'BACKGROUND' section of study protocol, (Hallensleben ND, Umans DS, Bouwense S, Verdonk RC, Besselink MG, Van Hoof JE, et al. The clinical course and diagnostic work-up of idiopathic acute pancreatitis, a post-hoc analysis of a prospective multicenter observational cohort. Digestive Disease Week; San Diego: Gastroenterology) there are no other studies to support that. Please comment on that.

Authors: The literature regarding this subject is scarce, yet it is still part of the recommendations as made by the IAP/APA evidence-based guidelines on management of acute pancreatitis. This is the best evidence available to make a substantiated assumption of the effects of this recommendation,

and thus, this study was chosen for this objective. The study is now published and the reference is updated.

3) 'In PICUS, biliary etiology is defined as either the signs of cholelithiasis, microlithiasis or biliary sludge on transabdominal ultrasonography, or transient elevation of the blood serum ALT level of more than twice the upper limit of normal at admission'.

However, not all patients with twice elevated ALT are having biliary etiology for acute pancreatitis. As per studies quoted in the 'Discussion' section, positive predictive value is 85% for an ALT > 150 U/L within 48 hours after onset of symptoms. This means still 15 percent can have other etiology. This requires further clarification by the authors.

Authors: We agree with dr. Somani that an elevated ALT level is not 100% correlated with biliary etiology. As elaborated on in our first rebuttal, we agree that comorbidity (e.g. hepatitis) could alter ALT levels, and the ALT levels may subsequently be wrongly interpreted as indicative of biliary pancreatitis. It is certainly not our intention to exclude these patients on the basis of having biliary pancreatitis. Therefore, in PICUS, we will only consider elevated ALT levels as indicative of biliary pancreatitis when they are transiently elevated at admission and normalize during admission, as is typical for biliary pancreatitis, and when there is no ALT altering comorbidity. We have clarified this matter in additional file 2 ("Relevant Definitions") and in the Discussion section.

4) 'In PICUS, all patients are required to undergo imaging after clinical recovery, i.e. transabdominal ultrasound or MRI/MRCP'.

However, biliary sludge can form after clinical recovery after acute pancreatitis or due to use of antibiotics like ceftriaxone.

This may overestimate the biliary etiology in the present study protocol.

Authors: We agree that timing of EUS may influence diagnostic yield, both positively as well as negatively. This issue was also mentioned by reviewer #3. Please see our first response to reviewer #3 above.